# The Effects of a Combination of Medical Cannabis, Melatonin, and Oxygen–Ozone Therapy on Glioblastoma Multiforme: A Case Report

**DOI:** 10.3390/reports6020022

**Published:** 2023-05-05

**Authors:** Marina Antonini, Cristina Aguzzi, Alessandro Fanelli, Andrea Frassineti, Laura Zeppa, Maria Beatrice Morelli, Gabriella Pastore, Massimo Nabissi, Margherita Luongo

**Affiliations:** 1Fondazione Policlinico Universitario Agostino Gemelli IRCCS, 00168 Roma, Italy; antonini.marina@hotmail.it; 2School of Pharmacy, University of Camerino, 62032 Camerino, Italy; cristina.aguzzi@unicam.it (C.A.); laura.zeppa@unicam.it (L.Z.); mariabeatrice.morelli@unicam.it (M.B.M.); 3Integrative Therapy Discovery Lab, School of Pharmacy, University of Camerino, 62032 Camerino, Italy; 4Clinical Research Institute Ecomedica, 50053 Empoli, Italy; alessandro.fanelli@ecomedica.it (A.F.); gabriella.pastore@ecomedica.it (G.P.); 5“Maria Guarino” Foundation-Amor No Profit Association, 80078 Pozzuoli, Italy; info@dottorfrassineti.it

**Keywords:** glioblastoma, cannabinoids, melatonin, oxygen–ozone therapy, chemotherapy, radiotherapy

## Abstract

Glioblastoma is the most aggressive malignant tumor overall and remains an incurable neoplasm with a median survival of 15 months. Since 2005, the gold standard treatment for glioblastoma has remained unchanged, and it is a common goal of the scientific community to work towards a better prognosis and improved survival for glioblastoma patients. Herein, we report a case of glioblastoma multiforme in a patient with a poor prognosis who, following partial removal of the neoplasm, refused conventional therapy consisting of a combination of radiotherapy and temozolomide-based chemotherapy due to personal serious side effects. The patient started an unconventional therapeutic path by alternating periods of oxygen–ozone therapy with concomitant administration of legal medical cannabis products (Bedrocan and Bedrolite) and melatonin. This approach resulted in a complete and durable remission of the disease and long survival. Indeed, the patient is still alive. The exceptional result obtained here encourages us to share and carefully investigate this unconventional treatment as a possible future direction in the management of glioblastoma.

## 1. Introduction

Glioblastoma (GBM) is the most aggressive malignant tumor overall and is considered a grade IV glioma, according to the World Health Organization (WHO) [1]. GMB accounted for 14.5% of all primary brain and other central nervous system (CNS) tumors, 48.6% of primary malignant brain tumors, and 57.3% of all gliomas [2]. To date, GBM remains an incurable tumor with a median survival of 15 months [3]. The current 1-year survival rate is 42.8%, while the 5-year survival rate is 7.2% [2].

The standard of care treatment for newly diagnosed GBM patients consists of surgical resection followed by radiation therapy (RT) and concurrent chemotherapy [4]. The chemotherapy of choice is temozolomide (TMZ), a nonspecific alkylating agent [5,6]. TMZ is a prodrug which undergoes rapid chemical conversion to the active monomethyl triazenoimidazole carboxamide (MTIC) at physiologic pH. The MTIC determines the methylation at the O-6 and N-7 positions of guanine and N-3 position of adenine, inducing DNA damage in tumor cells [7]. The inclusion of TMZ chemotherapy in treatment guidelines was based on a phase III clinical trial by the European Organization for Research and Treatment of Cancer (EORTC) and National Cancer Institute of Canada Clinical Trials Group (NCIC) [4]. In the EORTC–NCIC trials, patients were randomized to receive standard RT (60 Gy in 30 fractions over 6 weeks), with or without concomitant chemotherapy, that consisted of oral TMZ at a daily dose of 75 mg/m^2^ given 7 days per week for the entire duration of the radiotherapy. After a 4-week break, adjuvant TMZ (150–200 mg/m^2^) was administered for 5 days every 28 days up to a maximum of six cycles of monotherapy [8]. The study showed that the addition of concomitant and adjuvant TMZ to standard postoperative radiotherapy improved median survival and 2-year survival compared to postoperative radiotherapy alone [9,10]. Several drugs, such as bevacizumab or novel therapeutic targets such as poly (ADP-ribose), polymerase (PARP), epidermal growth factor receptor (EGFR), ataxia telangiectasia mutated (ATM), ataxia telangiectasia and rad3-related protein (ATR), and immune checkpoint, have been studied over the decades [11,12]. A number of clinical trials are ongoing to evaluate immune checkpoint inhibitors as a possible treatment for GMB [13,14]. However, no real clinical benefit has been found so far. In summary, the gold standard GBM treatment has remained the same since 2005. Hence, developments are required if we are to hope that future novel combination therapies will offer some much-needed survival improvement.

The effects of cannabinoids in hindering tumor progression have been deeply investigated at preclinical levels in vitro and in vivo, and two clinical trials further support the potential effects of phytocannabinoids in GBM [15,16,17,18].

The O_2_O_3_ therapy is widely used as an adjuvant therapeutic option in several pathological conditions characterized by chronic inflammatory processes and immune overactivation [19]. Several studies have underlined the relevant medical features of O_2_O_3_ therapy [19,20]; its efficacy could be related to the moderate oxidative stress modulation produced by the interaction of the ozone with biological components. In addition, the ozone, through oxidative preconditioning, defends tissues from reactive oxygen species (ROS)-related damage, supporting the antioxidant–prooxidant balance and the concomitant preservation of the redox cell state [21].

Melatonin (MLT), or N-acetyl-5-methoxytryptamine, has antioxidant effects too. It is an indoleamine that is synthesized and secreted by the pineal gland [22] and is involved in the regulation of biological rhythms and endocrine functions. The potential role of MLT in tumor treatment has been investigated through various studies [23,24,25,26], so it was included in this unconventional therapy.

In GBM, no previous studies have examined the relationship between MLT and cannabinoids in conjunction with O_2_O_3_ therapy. The preclinical evidence on the efficacy of CBD and O_2_O_3_ treatment was reported in human pancreatic cancer cells [27]. Furthermore, a previous study reviewed the use of ozone therapy in enhancing the action of chemotherapy and in reducing side effects [28].

Herein, we report the case of a GBM patient with a poor prognosis who declined the conventional adjuvant chemoradiotherapy, after partial surgical resection, due to personal serious side effects. The patient starts an uncommon treatment involving O_2_O_3_ therapy, legal medical cannabis, and MLT. Seven years after the diagnosis of GBM, the patient was considered totally cured. This exceptional result encouraged us to share and carefully investigate this unconventional treatment as a possible future direction in the management of GBM.

## 2. Case Report

In August 2016, in the emergency department, a 36-year-old woman, after 15 days of headache and unresponsiveness to non-steroidal anti-inflammatory drugs (NSAID), underwent a computed tomography scan (CT scan). The CT scan of the brain, performed with contrast medium, showed, in the left fronto-parietal area, two voluminous formations and inhomogeneous contiguous focalities of 52 × 41 mm and 40 × 20 mm, respectively, characterized by irregular peripheral impregnation and contextual necrotic–colliquative areas, with surrounding perilesional digitiform hypodensity, compression of the lateral ventricle, and contralateral shift of the midline structures (1 cm) (Figure 1).

On 24 August 2016, partial removal of the left frontal multicentric neoplasm was performed. The glial neoplasm consisted of a population of morphologically heterogeneous, highly proliferating cellular elements with MIB-1 labeling index of 70%, O-6-methylguanine-DNA methyltransferase (MGMT) methylated, and isocitrate dehydrogenase (NADP(+)) 1 (IDH-1) wild type. Upon histological examination, the patient was diagnosed with GMB and had an unfavorable prognosis.

Encephalon/Cranium CT scan with contrast medium performed postoperatively on 12 October 2016 showed an apparently reduced liqueur-like collection with associated wall impregnation. Posterior to the surgical cavity, there is an approximately 2 cm hyperdense solid nodule characterized by contrast-enhancing impregnation (Figure 2).

From October to November 2016, the patient was submitted to brain radiotherapy at the left fronto-parietal level (59.4 Gy delivered in 33 daily fractions of 1.8 Gy each) and chemotherapy with TMZ at the dose of 120 mg daily for 5 days. After the first cycle of chemotherapy, the patient refused further treatments with TMZ due to thrombocytopenia and personal marked side effects.

In November 2016, the patient started an integrated therapy consisting of oxygen–ozone (O_2_O_3_) therapy, MLT, and medical cannabis.

O_2_O_3_ therapy consisted of the rectal administration of a volume of 2.5 mL/kg of O_2_O_3_ (97% oxygen, 3% ozone), at a concentration of 80 µg/mL, for 2 daily administrations (interval of six hours) for 4 consecutive days per week for the first month. On day 5, a control blood count was carried out, and at the end of the month, a complete biochemistry. For the following 2 months, the treatment involved a daily administration for 4 consecutive days per week. At the end of the first 3 months of O_2_O_3_ therapy, the patient discontinued the treatment for 3 months. The therapy was then restarted and maintained until today, reducing the weekly sessions from four to two, and alternating between 3 months of treatment and 3 months off. MLT was initially administered per os at a dosage of 100 mg once a day and then increased by 100 mg every 4 days, up to a maximum of 2 g. Regarding medical cannabis, the patient assumed an initial dose of 50 mg of Bedrolite/4 times daily (CBD 9% and Δ9-THC 0.4%) and 100 mg of Bedrocan/4 times daily (Δ9-THC 22% and CBD < 1.0%). This then increased over the first 2 years to 1–5 g daily for both. From 2019, Bedrolite treatment was changed to 100 mg/4 times daily during the O_2_O_3_ pause phase and 50 mg/4 times daily during the O_2_O_3_ therapy.

During the follow-up, quarterly visits, clinical-instrumental checks, and monthly blood and biochemistry tests were carried out. At the end of the first cycle of O_2_O_3_ therapy (9 February 2017), the 2 cm residual tumor mass that was present 3 months before was no longer visible on the brain magnetic resonance imaging (MRI) (Figure 3), and the stable-negative tumor mass was confirmed with the MRI on 11 October 2022 (Figure 4).

MRIs were done in subsequent periods (12 July 2021 and 11 October 2022), confirming that the morphological picture remained stable and that the patient was considered cured (Figure 5).

## 3. Discussion

We report the case of a patient with GBM who, after partial excision of the neoplasm and radiotherapy, refused conventional chemotherapy due to personal intolerable side effects. The patient started a supportive therapy that included O_2_O_3_ therapy, MLT, and medical cannabis.

After the first report on the antineoplastic activity of cannabinoids in 1975 [15], several investigations into phytocannabinoids have uncovered a range of interesting findings about their mechanisms of action, the pathways involved in their effects, and their safety and tolerability. Phytocannabinoids are a group of bioactive compounds derived from *Cannabis sativa* plant, including Δ9-tetrahydrocannabinol (Δ9-THC) and cannabidiol (CBD) [29,30]. Cannabinoids interact with cannabinoid receptor 1 (CB1) and cannabinoid receptor 2 (CB2), acting as agonists or inverse agonists [29]. Cannabinoids can also target other receptors, including G-protein-coupled receptors (GPCRs), GPR12, GPR18, GPR35, GPR55, GPR119, opioid and serotonin receptors, and transient receptors potential channels (TRPV1, TRPV2, TRPA1, etc.) [31,32,33]. The analysis of GBM cell lines, ex vivo primary tumor cells, and GBM tissue biopsies showed that GBM tumors express both cannabinoid receptors CB1 and CB2, with high-grade tumors expressing high levels of CB2. This dysregulated expression of cannabinoid receptors in GBM led to the hypothesis that phytocannabinoids present in medical cannabis may be effective for therapeutical approaches [34,35]. On this basis, several in vitro and in vivo studies have investigated different phytocannabinoids for anti-tumor activity in glioma [31]. A phase I clinical trial (part 1 and part 2) analyzed the safety and tolerability of nabiximols, a cannabinoids-based oromucosal spray, administered in combination with dose-intense TMZ (DIT) in patients with recurrent GBM, as well as patients’ progression-free survival at 6 months and overall survival (OS) at 1 year. The study showed increased efficacy in patients with adjuvant nabiximols (survival at 1 year was 83% and and 44% for nabiximols- and placebo-treated patients, respectively), despite the limitation of the small sample size [16]. Furthermore, a phase II randomized clinical trial assessed the tolerability of two different ratios of medicinal cannabis in patients with high-grade gliomas, showing that its administration is safe, well-tolerated, and capable of providing symptomatic relief to patients [17]. Moreover, because of a systematic review of the literature on clinical and experimental trials on the antitumor effects of cannabinoids, Velasco and colleagues claimed that cannabinoids impair tumor progression at various levels. The inhibition of tumor growth derives from different mechanisms consisting of the induction of cancer cell death by apoptosis, the inhibition of cancer cell proliferation, the impairment of tumor angiogenesis, and the blocking of invasion and metastasis [35].

Along with cannabinoids, MLT has been employed in this therapy. It stimulates the production of cytokines, in particular interleukins IL-2, IL-6, IL-12, and increases T helper immune responses [22]. Furthermore, MLT has retinal functions and antioxidant actions that contribute to its immunoenhancing effects, indirectly reducing nitric oxide formation and facilitating a decrease in inflammatory response [22,23]. A meta-analysis conducted in recent years evaluating 5057 articles highlighted how MLT exerts a positive influence in tumor therapeutic strategies, exhibiting improvements in tumor remission and OS rate and and a reduction in chemotherapy-induced side effects, showing a significant effect in preventing, treating, and delaying tumor development [24]. Similarly, the work of Fernandez and colleagues showed that reduced MLT levels are associated with an increased cancer risk, suggesting the oncostatic action of this hormone [25]. However, a systematic review and meta-analysis evidenced that MLT did not improve the quality of life, sleep quality, fatigue, pain or stomatitis severity among patients with cancer [36].

It should be considered that the different methods of administration, durations, and cancer types were the main sources of heterogeneity in MLT effects, and further clinical trials are certainly needed.

Regarding O_2_O_3_ therapy in cancer, no data are available on cancer patients. A previous study reviewed its use in enhancing the action of chemotherapy and in reducing side-effects [19,20], and its potential effect in reducing pain in chronic disease [28,37].

## 4. Conclusions

Previous preclinical and clinical studies have investigated the combination of THC and CBD associated with TMZ in GBM [15,16,17]. In the case reported in this paper, it was not possible to evaluate the combination with TMZ because the patient refused this therapy due to personal marked side effects.

The use of O_2_O_3_ therapy with concomitant administration of legal medical cannabis products and MLT contributed to the patient’s complete recovery, but to date, the available scientific literature is still insufficient to confirm our hypothesis, especially considering that most clinical trials have been performed on individual substances and not on the therapy as a whole. Further clinical studies are certainly needed to better verify whether this unconventional therapy can be considered as a valid supportive treatment for GBM and other tumors.

It is important to note that the patient remained without disease progression during the time of the therapy and did not develop side effects during treatments.

## Figures and Tables

**Figure 1 reports-06-00022-f001:**
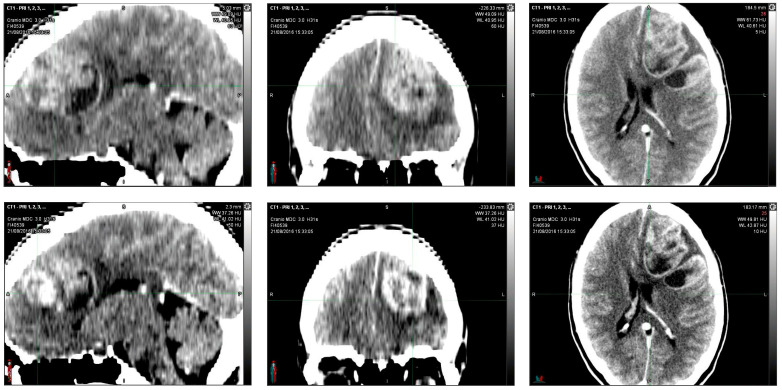
Computerized tomography (CT) scan with contrast medium performed at initial diagnosis (21 August 2016) showed, in the left fronto-parietal area, two voluminous formations and inhomogeneous contiguous focalities of 52 × 41 mm and 40 × 20 mm, respectively, characterized by irregular peripheral impregnation and contextual necrotic-colliquative areas, with surrounding perilesional digitiform hypodensity, compression of the lateral ventricle, and contralateral shift of the midline structures (1 cm).

**Figure 2 reports-06-00022-f002:**
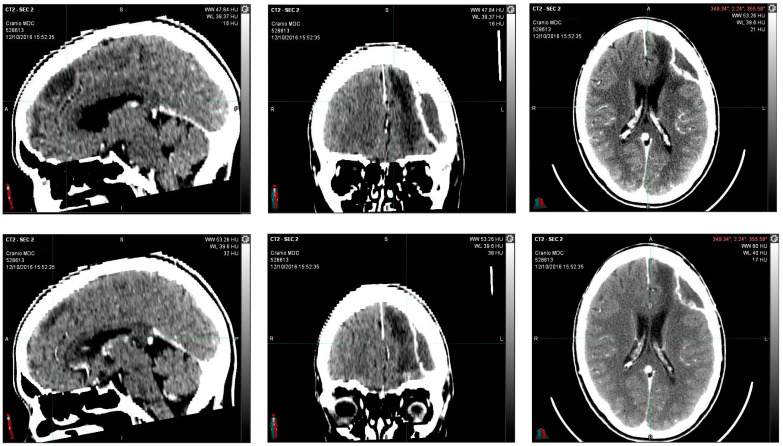
12 October 2016, instrumental examination after partial removal of left frontal multicentric neoplasm. The CT scan images with contrast medium showed an apparently reduced liqueur-like collection with associated wall impregnation. Posterior to the surgical cavity, there is an approximately 2 cm hyperdense solid nodule characterized by contrast-enhancing impregnation.

**Figure 3 reports-06-00022-f003:**
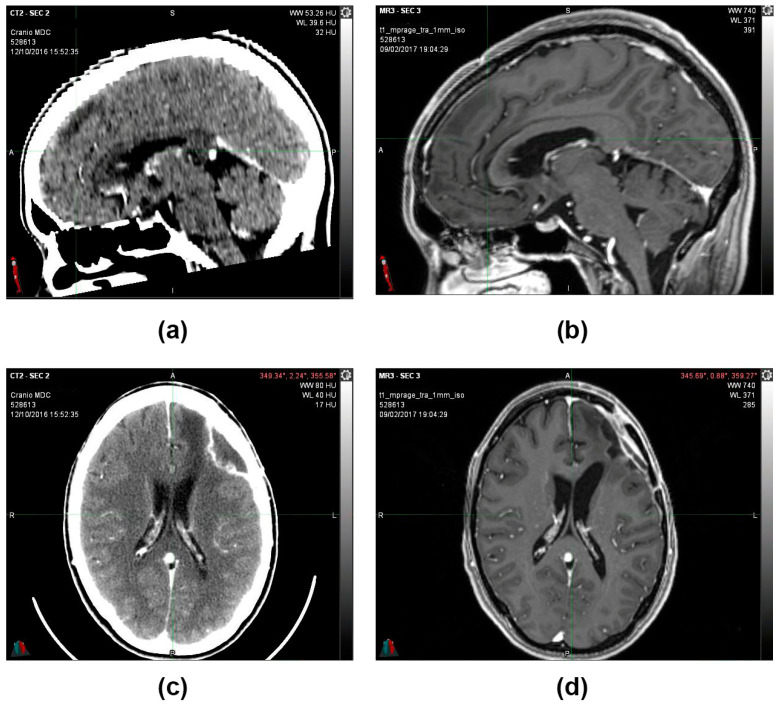
(**a**,**c**) 12 October 2016, CT scan image with contrast medium. Posterior to the surgical cavity, an approximately 2 cm hyperdense solid nodule is present (**b**) 9 February 2017, MRI T1 sequence with contrast medium. After the first cycle of O_2_O_3_ therapy, the 2 cm residual tumor mass that was present 3 months before was no longer visible (**d**) 9 February 2017, MRI image sequence in Fluid-attenuated inversion recovery (FLAIR).

**Figure 4 reports-06-00022-f004:**
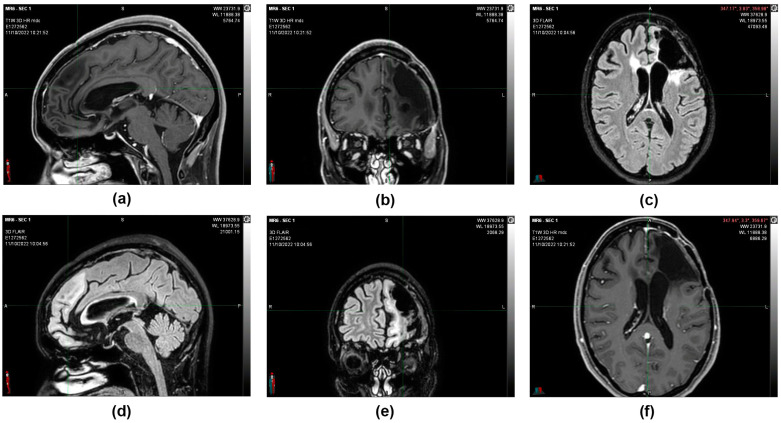
11 October 2022, instrumental examination, stable negative picture. (**a**,**b**,**f**) MRI images, T1 sequence with contrast medium. (**c**–**e**) MRI images, sequence in FLAIR. The morphological picture remained stationary—the T2 hyperintensity surrounding the known left frontal malacic cavity extended to the knee and right root of the corpus callosum and in the left semioval center and was without mass effect and not modified in the contrastographic phases. Non-focality of new onset at the remaining levels. In the spongiosa of the left frontal bone in the median region and on the profile of the operculum, there are two areas of altered signal with contrastographic impregnation, both of which are stationary. The median structures are on an axis. The ventricular cavities and subarachnoid spaces are overlapping in size and morphology.

**Figure 5 reports-06-00022-f005:**
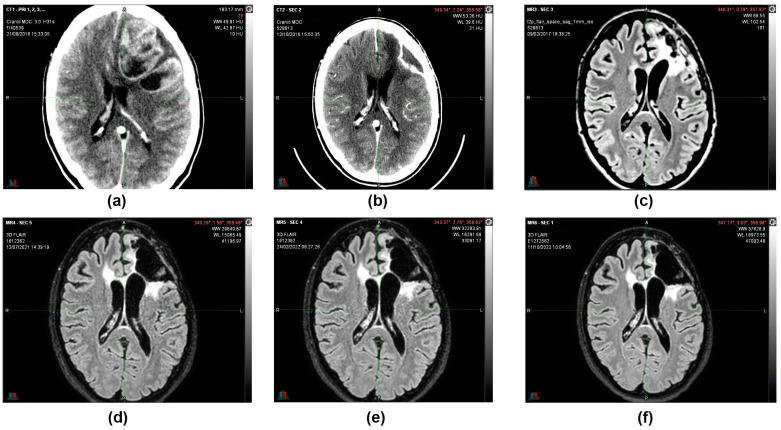
(**a**) 21 August 2016, CT image with contrast medium before surgery. (**b**) 12 October 2016, CT image with contrast medium after partial removal of left frontal multicentric neoplasm. (**c**) 9 February 2017, MRI image, T1 sequence with contrast medium after the first cycle of oxygen-ozone therapy. (**d**) 12 July 2021, MRI image sequence in FLAIR. (**e**) 24 February 2022, MRI image sequence in FLAIR. (**f**) 11 October 2022, MRI image sequence in FLAIR. Following a total of 5 years after the partial resection surgery, the morphological picture remained stable, and the patient was considered cured.

## Data Availability

The data presented in this study are available on request from the corresponding author. The data are not publicly available due to their containing information that could compromise the privacy of patient subject of this case report.

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
