# Peer review of "The Effects of a Combination of Medical Cannabis, Melatonin, and Oxygen–Ozone Therapy on Glioblastoma Multiforme: A Case Report"

_reports, 2023, doi:10.3390/reports6020022_

Round 1
Reviewer 1 Report
This intriguing case report concerns the life prolongation of a young female patient baring GBM grade IV, a devastating brain tumor with dismal prognosis. Since the conventional post-surgery radiotherapy was followed by only one cycle of temozolomide chemotherapy –discontinued owing to marked side effects–, the patient’s exceptional clinical course (still alive and tumor-free 6 years after the diagnosis) was attributed to an unconventional therapeutic regimen consisting of alternating periods of oxygen-ozone therapy with concomitant administration of medical phytocannabinoids and melatonin.
1. Introduction & Discussion: The Introduction is disproportionally lengthy compared to the Discussion. Many of the issues addressed in the Introduction (e.g., lines 70-121) appear as more appropriate for the Discussion section.
1. Case Report (lines 162-176): Who introduced and supervised the chosen therapeutic protocol? Regarding its duration, is it expected to be carried out for life? Were its details (e.g., volumes, concentrations, dosages, and alternating administration scheme of O2O3, melatonin and canabinnois) chosen arbitrarily, or are they based on some pre-existing published data? Please provide relevant literature references, if available.
2. Discussion: Can the authors propose some potential metabolic pathways and channels on the cellular level, where the three chosen therapeutic substances could be intersecting to produce this prolonged tumor remission?
3. Please define at first appearance the abbreviation ROS (line 238).
4. The use of English is on a good level, with only minor occasional corrections needed.
Author Response
Camerino, 17 April 2023
REBUTTAL LETTER
Dear Editor,
Please find enclosed the electronic version of the revised case report entitled “The effects of a combination with medical cannabis, melatonin and oxygen-ozone therapy in glioblastoma multiforme. A case report.” by Marina Antonini, Cristina Aguzzi, Alessandro Fanelli, Andrea Frassineti, Laura Zeppa, Maria Beatrice Morelli, Gabriella Pastore, Massimo Nabissi and Margherita Luongo.
We thank the reviewers for carefully reading the manuscript and for constructive remarks. We have taken the comments on board to improve and clarify the manuscript. Please find below a detailed point-by-point response to all comments.
Reviewer 1:
Comments and Suggestions for Authors
This intriguing case report concerns the life prolongation of a young female patient baring GBM grade IV, a devastating brain tumor with dismal prognosis. Since the conventional post-surgery radiotherapy was followed by only one cycle of temozolomide chemotherapy –discontinued owing to marked side effects–, the patient’s exceptional clinical course (still alive and tumor-free 6 years after the diagnosis) was attributed to an unconventional therapeutic regimen consisting of alternating periods of oxygen-ozone therapy with concomitant administration of medical phytocannabinoids and melatonin.
- Introduction & Discussion: The Introduction is disproportionally lengthy compared to the Discussion. Many of the issues addressed in the Introduction (e.g., lines 70-121) appear as more appropriate for the Discussion section.
- Thanks for the comments. The paper was modified as suggested, improving the discussion section with respect to the Introduction section. References were updated.
- Case Report (lines 162-176): Who introduced and supervised the chosen therapeutic protocol? Regarding its duration, is it expected to be carried out for life? Were its details (e.g., volumes, concentrations, dosages, and alternating administration scheme of O2O3, melatonin and cannabinoids) chosen arbitrarily, or are they based on some pre-existing published data? Please provide relevant literature references, if available.
- The therapies were performed at the medical office of the Medical Foundation Maria Guarino Onlus. The multitherapy was designed and performed by Dr. Margherita Luongo, who reported this patient's case and provided the data for this case report.
- The patients terminated the therapy at the end of 2022.
- The doses and administration scheme of O2O3, were chosen on the basis of the Medical Foundation's experiences with patients, following some indications reported in the reference [28]. No literature references are available regarding this combination therapy.
- Discussion: Can the authors propose some potential metabolic pathways and channels on the cellular level, where the three chosen therapeutic substances could be intersecting to produce this prolonged tumor remission?
The potential pathways throw which medical cannabis, melatonin and oxygen-ozone therapy act, are indicated (lines 228-230; 232-236; 71-75). However, little information was published on potential pathways interaction, since the only preclinical data in cancer cells that was published is related to the combination of Cannabidiol and Oxygen-ozone therapy (ref.27), which suggested that CBD and O2O3 synergically reduced cell viability and induced cell death in human pancreatic cancer cell lines.
- Please define at first appearance the abbreviation ROS (line 238).
The abbreviation ROS (Reactive Oxygen Species) was defined in the text (line 73).
- The use of English is on a good level, with only minor occasional corrections needed.
As suggested by the reviewer, we corrected the manuscript to eliminate errors.

Reviewer 2 Report
This is a case report of a GBM patient. As a case report, it is well written and informative. Scientifically, the case report has little value, which is in the nature of things. This report could serve to determine the importance of ozone therapy (cannabis and melatonin) in GBM.
Author Response
Camerino, 17 April 2023
REBUTTAL LETTER
Dear Editor,
Please find enclosed the electronic version of the revised case report entitled “The effects of a combination with medical cannabis, melatonin and oxygen-ozone therapy in glioblastoma multiforme. A case report.” by Marina Antonini, Cristina Aguzzi, Alessandro Fanelli, Andrea Frassineti, Laura Zeppa, Maria Beatrice Morelli, Gabriella Pastore, Massimo Nabissi and Margherita Luongo.
We thank the reviewers for their careful reading of the manuscript and constructive remarks. We have taken the comments on board to improve and clarify the manuscript. Please find below a detailed point-by-point response to all comments.
Reviewer 2
This is a case report of a GBM patient. As a case report, it is well written and informative. Scientifically, the case report has little value, which is in the nature of things. This report could serve to determine the importance of ozone therapy (cannabis and melatonin) in GBM.
We thank the reviewer for the positive comments.
Reviewer 3 Report
In this paper, the author present a quite interesting case report of a young patient suffering from GBM, who refused standard therapy and remained stable 6 years after surgery with a combination therapy of medical cannabis, melatonin and oxygen-ozone. Although the survival time is quite impressive, to my opinion there need to be some things clarified.
1. the authors claim an initial life prognosis of 7 month- please explain
2. the neuropathology report is from 2016 according to the older classification- according to the actual classification: is it an IDH wild-type glioblastoma or would the diagnosis have been changed? what about MGMT?
3. standard imaging to interpret the result of surgery regarding extend of resection is MRI-even in 2016- to my opinion it is not possible to state if there is residual Tumor or not and how it developed over time
Therefore, two of the main statements of this report have to be clarified.
Minor aspects: a) how much radiation therapy did the patient receive? b) Please change the word "study" into "case report". c) In the introduction it sounds, that concomitant radio-TMZ chemotherapy is inevitably connected to maximal side effects, which is not the case. d) Since 2005 there are also more upcoming therapy regimes- CCNU? TTF? Immunotherapy?
Author Response
Camerino, 17 April 2023
REBUTTAL LETTER
Dear Editor,
Please find enclosed the electronic version of the revised case report entitled “The effects of a combination with medical cannabis, melatonin and oxygen-ozone therapy in glioblastoma multiforme. A case report.” by Marina Antonini, Cristina Aguzzi, Alessandro Fanelli, Andrea Frassineti, Laura Zeppa, Maria Beatrice Morelli, Gabriella Pastore, Massimo Nabissi and Margherita Luongo.
We thank the reviewers for carefully reading the manuscript and for the constructive remarks. We have taken the comments on board to improve and clarify the manuscript. Please find below a detailed point-by-point response to all comments.
Reviewer 3
In this paper, the author presents a quite interesting case report of a young patient suffering from GBM, who refused standard therapy and remained stable 6 years after surgery with a combination therapy of medical cannabis, melatonin and oxygen-ozone. Although the survival time is quite impressive, to my opinion there need to be some things clarified.
- the authors claim an initial life prognosis of 7 month- please explain.
Since the life prognosis was reported by the patient, we modified the sentence to poor prognosis according to WHO and the National Brain Tumor Society. Indeed, GBM is the most common malignant primary brain tumor in adults and prognosis is generally very poor, with a median overall survival (OS) of less than 15 months, and a 5-year OS rate of only 10%, even when aggressively treated.
The neuropathology report is from 2016 according to the older classification- according to the actual classification: is it an IDH wild-type glioblastoma or would the diagnosis have been changed? what about MGMT?
The patient was MGMT methylated and IDH-1 wild type. The information was added to the text.
- standard imaging to interpret the result of surgery regarding extend of resection is MRI-even in 2016- to my opinion it is not possible to state if there is residual Tumor or not and how it developed over time.
In 2016 the patient was subject to CT only; no MRI were performed before 2017. The first MRI was performed in February 2017, as reported in Figure 3.
Therefore, two of the main statements of this report have to be clarified.
Minor aspects: a) how much radiation therapy did the patient receive? b) Please change the word "study" into "case report". c) In the introduction it sounds, that concomitant radio-TMZ chemotherapy is inevitably connected to maximal side effects, which is not the case. d) Since 2005 there are also more upcoming therapy regimes- CCNU? TTF? Immunotherapy?
- The patient received 59.4 Gy delivered in 33 daily fractions of 1.8 Gy each (line 130)
- In all cases in which the term "study" has been used, reference is made to clinical trials or scientific articles, the only case in which reference is made to our case report is line 266. We replace the word “study” into “therapy”, as rightly highlighted by the reviewer.
- The TMZ-induced side effects sentence was reported to explain why the patient refused chemotherapy (lines 25, 88-89, 133, 198-199, 256-257). With this sentence, we do not want to affirm that the radio-chemotherapy with TMZ combination always is inevitably linked to maximal side effects. We modified the text by adding the sentence “personal side effects”, to underline that not all the patients under this therapy showed side effects.
- In response to the reviewer, we added the sentence “Several drugs such as bevacizumab or novel therapeutic targets like poly (ADP-ribose) polymerase (PARP), epidermal growth factor receptor (EGFR), ataxia telangiectasia mutated (ATM), ataxia telangiectasia and rad3-related protein (ATR), and immune checkpoint, have been studied over the decades” (lines 56-60).
Round 2
Reviewer 1 Report
The authors have addressed the review’s remarks and the revised manuscript has improved. However, the last two of the article’s figures (i.e., Figure 4 and Figure 5) remain uncited in the main text of the manuscript. They either have to be mentioned in the text, or removed.
Author Response
We apologize for the omission. Text referring to Figure 4 and 5 have been added
Reviewer 3 Report
no further comments- thank you for the revision
Author Response
Thanks for the positive evaluation.